# A Susceptible Scion Reduces Rootstock Tolerance to *Ralstonia solanacearum* in Grafted Eggplant

**Chaokun Huang** [1,2,†] , **Yuexia Wang** [1,†], **Yanjuan Yang** [1], **Chuan Zhong** [1], **Michitaka Notaguchi** [2] **and Wenjin Yu** [1,*]

[1]   Horticultural Science, Agricultural College of Guangxi University, Nanning 530004, China; hck5632@icloud.com (C.H.); wangyueya33@163.com (Y.W.); yjyang85@126.com (Y.Y.); zhongchuan11@hotmail.com (C.Z.)
[2]   Bioscience and Biotechnology Center, Nagoya University, Nagoya 4648601, Japan; notaguchi.michitaka@b.mbox.nagoya-u.ac.jp
*   Correspondence: yuwjin@gxu.edu.cn
†   These authors contributed equally to this work.

**Abstract:** The bacterial wilt pathogen (*Ralstonia solanacearum*) is a highly pathogenic soil-borne bacterium that invades the vascular system of a host plant leading to plant wilting and death. In agricultural systems, tolerant rootstocks are usually used to enhance disease resistance and tolerance in crop plants to soil-borne pathogens. Here, two distinct eggplant cultivars with different tolerances to *R. solanacearum* infection, the disease-tolerant cultivar 'S21' and the disease-susceptible cultivar 'Rf', were used to investigate if scion tolerance level can affect tolerance of rootstock upon an infection of the same pathogen. Three scion/rootstock grafted combinations were considered: Rf/S21, S21/S21, and Rf/Rf. Plants that resulted from the combination Rf/S21, composed of the susceptible scion grafts, showed weak tolerance to *R. solanacearum* infection, and exhibited the poorest growth compared to the tolerant scion grafts (S21/S21). As expected, the combination Rf/Rf showed the lowest level of disease tolerance. Furthermore, a high level of exopolysaccharides (EPSs) and cell wall degrading enzymes (CWDEs) were detected in susceptible scion grafts. These factors are involved in plant growth inhibition due to blocking transport between scion and rootstock and damage of vascular tissues in the plant. A high level of reactive oxygen species (ROS) and active oxygen scavenging enzymes were also detected in susceptible scion grafts. Excess accumulation of these substances harms the dynamic balance in plant vascular systems. These results indicated that the use of a susceptible scion in scion/rootstock eggplant grafts contributed to a reduction in rootstock tolerance to *Ralstonia solanacearum*.

**Keywords:** bacterial wilt; *Solanum melongena*; susceptible; tolerance; exopolysaccharides; cell wall degrading enzymes

## 1. Introduction

Bacterial wilt is a soil-borne disease of eggplant (*Solanum melongena*) caused by *Ralstonia solanacearum E.F. Smith*, which usually occurs in tropical, subtropical, and temperate regions (Smith, 1896). The bacterial wilt pathogen enters the host plant through the roots and invades the vascular system [1,2]. *R. solanacearum* infects xylem parenchyma and stimulates cells to form invasive compounds in the vicinity of the vascular tissues, which are then released into the vessels and sieve tubes. The pathogen rapidly proliferates and spreads in vascular tissues, causing obstruction of the xylem and phloem [3]. *R. solanacearum* is also similar to most plant pathogenic bacteria that can secrete various

toxic proteins that induce a hypersensitive response in the host plant and activate a complex plant defense network [4].

Studies have shown that *R. solanacearum* can produce a large number of exopolysaccharides (EPSs) in the vascular tissues, which block xylem and phloem [5]. EPSs hinder water and nutrient transport in the plant's vascular system, resulting in wilting symptoms, and therefore EPSs are an important pathogenicity factor of bacterial wilt disease [6,7]. *R. solanacearum* secretes cell wall degrading enzymes (CWDEs) during host infection, including cellulase (Cx), pectin methylgalacturonase (PMG), and pectin methyl transelimination enzyme (PMTE), which have been studied for their potential in promoting pathogenesis and causing disease symptoms [8–10]. In general, CWDEs produced by plant pathogens are considered to be important pathogenicity factors. Significant work has been published on the role of CWDEs during plant pathogen infection, and studies show that the accumulation of CWDSs will accelerate the decline of plant cells [11,12].

Once host plants are infected with *R. solanacearum*, pathogens cannot be directly inhibited or eliminated, therefore plants induce and excrete defense factors including phytoalexin and reactive oxygen species (ROS) that enhance defense against pathogens [13]. Superoxide anion ($O_2^-$) and hydrogen peroxide ($H_2O_2$) are the major forms of ROS. These molecules are highly reactive and toxic and can lead to the oxidative destruction of cells. The rapid accumulation of plant ROS at the pathogen attack site, a phenomenon called oxidative burst, is directly toxic to pathogens and can lead to a hypersensitive response that results in a zone of host cell death, which prevents the further spread of biotrophic pathogens [14,15]. However, a greater degree of *R. solanacearum* stress on the plant tissues results in the excess accumulation of ROS, which increases the burden of the active oxygen-scavenging system in plants [16]. Active oxygen-scavenging systems in plants mainly include superoxide dismutase (SOD), catalase (CAT), peroxidase (POD), and ascorbate peroxidase (APX) [17,18]. The main role of SOD is to remove the $O_2^-$ present in the cytoplasm, chloroplasts, and peroxisomes to prevent oxygen radicals from damaging the cell membrane system [19]. CAT and POD facilitate $H_2O_2$ enzymatic degradation to prevent an over-accumulation of more toxic $OH^-$ [20]. APX is also a $H_2O_2$ scavenger that uses the APX-AsA cycle to reduce damage from $H_2O_2$ in plant cells [21,22]. These enzymes, to a certain extent, have become the protective system for plants facing environmental stress, maintaining the balance of ROS metabolism within the cells and protecting the membrane structure. However, plant cells will also be damaged if the excess accumulation of ROS exceeds the ability of the active oxygen scavenging enzyme system [23].

At present, the use of tolerant rootstocks for grafting eggplant varieties is the most effective approach to control bacterial wilt disease [24]. There are complex interactions between the rootstock and scion during the period of initiation and the completion of healing in grafts [25]. Investigations on the interaction of rootstock and scion have mostly been focused on physiological and biochemical characteristics regarding the mechanisms involved in graft compatibility, nutrient and water uptake, assimilation and translocation of solutes, and the influence of the rootstock on the main physiological processes of the scion [26]. Studies have shown that rootstocks not only affect the nutritional status, water metabolism, and hormonal signaling of the scion, but they can also increase the resistance and tolerance of the scion to disease and environment stresses, resulting in a higher yield [27,28]. Although a disease-tolerant rootstock can generally improve the tolerance of a susceptible scion [24], the disease tolerance of grafted plants is generally lower than that of self-rooted plants [29,30], which suggests that a susceptible scion may also influence the tolerance level of the rootstock. However, to our knowledge, there are no studies to determine if the disease tolerance level of a scion directly affects rootstock tolerance to a specific disease.

In this study, different grafted combinations of eggplant were established to determine if they affect plant growth and abate the bacterial wilt tolerance of a rootstock upon inoculation with *R. solanacearum*.

## 2. Materials and Methods

### 2.1. Plant Material and Ralstonia Solanacearum

Eggplant genotypes 'Qiezhen No. 21' (S21, high tolerance to bacterial wilt), bred by Guangxi University, Nanning, China, and "Zichang eggplant No. 1" (Rf, high susceptibility to bacterial wilt), a commercial variety, were used to establish the experiment. The grafted scion/rootstock combinations Rf/S21, S21/S21, and Rf/Rf were constructed as the experimental plants.

*R. solanacearum* (bacterial wilt pathogen) was provided by the Plant Pathology Laboratory of the Agricultural College at Guangxi University. The bacterium was mixed with a 30% glycerol solution and stored at −80 °C. Before inoculation, metal rods were dipped into the stored *R. solanacearum* and were used to draw several lines on sterile nutrient agar (NA) medium. The stored pathogens were transferred to the medium and were activated and incubated in a 28 °C incubator. After 2 d of incubation, media with no contamination and with good bacterial growth (a single white colony, strongly pathogenic in subsequent experiments) (Figure 1) were selected as inocula.

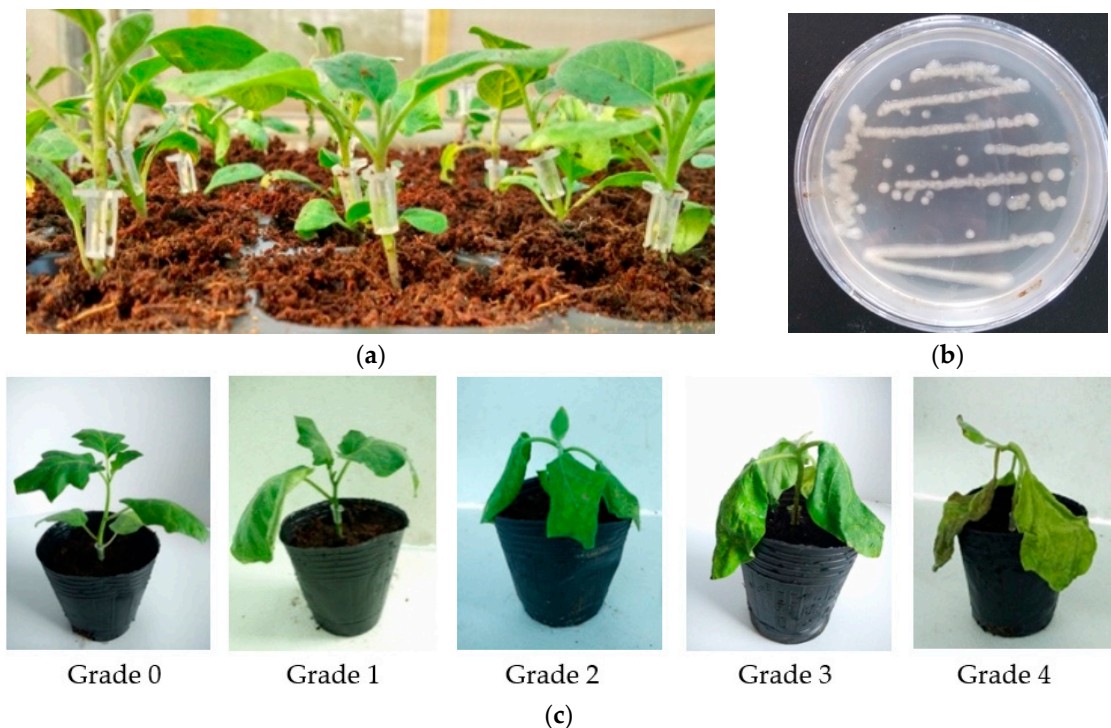

**Figure 1.** (**a**) Grafting seedlings with C-type annular tubes. (**b**) Pathogenic *R.solanacearum* in NA medium. (**c**) Disease classification standard for the plant.

### 2.2. Grafting Method, Grafted Plant Cultivation and R. solanacearum Inoculum

Eggplant seeds of each graft combination and a non-grafted Rf control were soaked in warm water (50 °C) and sown in 50-hole trays with commercial plant cultivation substrate to promote germination. When seedlings exhibited two true leaves, they were grafted with C-type annular tubes (Figure 1) using the close joining method [31]. The grafted seedlings were then kept in a growth chamber under controlled conditions (24–28 °C, 90–95% relative humidity, darkness) for 12 h. Further, to promote graft healing, the temperature of the growth chamber was changed to 20–22 °C for 3 d, then seedlings were moved to the greenhouse (26–28 °C). After 15 d of growth under greenhouse conditions, the grafts were transplanted into larger pots containing substrate. Physiological parameters (see in detail below) were measured during this period. When the grafted plants exhibited 4–5 true leaves, *R. solanacearum* inoculum, diluted in water to a concentration of $OD_{600} = 0.5$, was applied using the root damage infused inoculum method [32]. After inoculation, seedlings were incubated in a growth chamber for

28 d under a 14 h photoperiod with 30 °C day/25 °C night and 70% relative humidity. Non-inoculated plants of each graft combination were included and otherwise handled the same way. Prepare enough experimental plants according to the experimental arrangement (Table S1). Containers were watered and fertilized twice a week from seed sowing to the end of the experiment, approximately 2 to 3 months.

### 2.3. Disease Classification and Disease-Tolerance Evaluation

To identify disease symptoms, plants of the graft combinations and the non-grafted control were analyzed at 28 d post-inoculation (DPI) with *R. solanacearum*. Twenty plants were evaluated per grafted combination and control and repeated three times (80 plants per repetition, for a total of 240 plants, were considered; Table S1). The level of disease tolerance was evaluated at 28 DPI following the procedure described by Liu et al. [33]. Five different levels of plant tolerance to *R. solanacearum* infection were defined: Grade 0: no obvious symptoms; Grade 1: one leaf wilting; Grade 2: 2–3 leaves wilting; Grade 3: four leaves wilting; Grade 4: whole seedling wilting or death (Figure 1). The parameters, incidence rate (IR) and disease index (DI) were determined according to the mathematical expressions below. Incidence rate (IR) = number of infected plants/number of inoculated plants. Disease index (DI) = $\sum$ (disease grade × number of infected plants)/(the highest disease grade × number of inoculated plants) × 100. Five levels of plant tolerance to bacterial wilt were considered (based on the DI parameter): I: immune (DI = 0); HT: highly tolerant ($0 < DI \leq 15$); T: tolerant ($15 < DI \leq 30$); MT: moderately tolerant ($30 < DI \leq 45$); S: susceptible ($45 < DI \leq 60$); HS: highly susceptible (DI > 60) [9]. The experiment was a completely randomized design with three repetitions, analysis of variance (ANOVA) was used to test for significance, and significant differences ($P < 0.05$) between treatments were determined using Tukey's test.

### 2.4. Measurement of Plant Growth Situation

Ten plants from each graft combination were randomly selected after disease-tolerance evaluation (30 plants per experimental repetition, a total of 90 plants upon 3 repetitions; Table S1) at 28 DPI. The biomass that reflects plant growth was measured, including plant height (g), stem diameter (cm), root length (cm), and surface area ($cm^2$), and dry and fresh weight (g) of the plants. Plant height was considered as length between the base of the stem and the end of the apical meristem. Plant height was measured by a straightedge ruler. The stem diameter of the scion was measured approximately 0.5 cm above the graft interface and stem diameter of the rootstock was measured approximately 0.5 cm below the graft interface, both using a Vernier caliper. After washing roots to remove media, root length and surface area were measured using a Founder Z2400 scanner and analyzed with WinRHIZO 2009c software. Fresh weights of plant stems and roots were measured with an electronic Mettler Toledo balance. For dry weight determination, seedlings were placed in a ETTAS ON-300S drying oven, at 105 °C for 15 min for enzyme inactivation, dried at 45 °C until a constant weight, and then measured with the electronic balance. To calculate normalized values in this experiment, the biomass measurement values at 28 DPI minus their average value at 0 DPI (average from 10 plants per grafted combination), were calculated.

### 2.5. Determination of EPS Content and Cell Wall Degrading Enzyme (CWDE) Activity of Grafted Plants

Samples were collected from grafted plants at 0, 7, 14, and 28 DPI with *R. solanacearum*. Ten plants were selected from each grafted combination at each experimental timepoint (30 plants per grafted combination at each time point, for a total of 360 plants upon 3 repetitions; Table S1). Samples taken from stems (0.2 g) and roots were individually ground well with a mortar and pestle and diluted with 10 mL distilled water. After homogenization, samples were boiled for 30 min, centrifuged at 12,000× *g* for 10 min at 25 °C, and supernatants collected into a 25 mL bottle. These supernatants were then used to measure EPS content following the phenol-sulfuric acid method described by Cao et al. [34].

Briefly, 0.5 mL of supernatant solution was mixed with 1.5 mL of distilled water and after mixing well, 1.0 mL of phenol solution (90 g·L$^{-1}$) and 5.0 mL concentrated sulfuric acid were added. After 30 min of incubation at room temperature, absorbance was measured at a 485 nm wavelength.

To proceed with CWDE activity measurements, extraction buffer was prepared by dissolving 11.78 g NaCl in a 250 mL sodium acetate–acetic acid buffer solution (0.05 M, pH 5.5). Afterwards, samples (0.2 g) taken from stems and roots were individually thoroughly ground with a mortar and pestle and diluted with 1.6 mL extraction buffer in an ice bath. Then, centrifugation was performed at 12,000× *g* and 4 °C for 20 min, and supernatants (extracted solution) were collected for measuring CWDE activity [35].

To measure the cellulase (Cx) activity, the protocol described by Li et al. [35] was followed. Briefly, 0.1 mL of extracted solution was added to 0.2 mL carboxymethylcellulose sodium (CMC) solution (0.6% *v/v* concentration), with a 30 min incubation in a 50 °C water bath. Then, 1.5 mL 3,5-dinitrosalicylic acid (DNS) solution (6.3 g/L) was immediately added to stop the reaction. The absorbance was measured at a 485 nm wavelength.

To determine the pectin methyl-galacturonase (PMG) activity, the protocol previously described by Li et al. [35] was followed. Briefly, 0.1 mL of extracted solution was added to a solution consisting of 0.1 mL pectin solution (0.25% *v/v* concentration) and 0.3 mL acetate buffer (0.05 M, pH 5.5). Afterwards, 1.5 mL DNS solution (6.3 g/L) was added to stop the reaction. The absorbance was measured at a 540 nm wavelength.

The method used to determine the pectin methyl trans elimination enzyme (PMTE) activity followed the protocol described by Li et al. [35], consisting of addition of 0.1 mL of extracted solution to the previous prepared mixture of 0.1 mL pectin solution (0.25% concentration, containing 0.6 mmol·L$^{-1}$ of CaCl$_2$·2H$_2$O) and 0.3 mL glycine–sodium hydroxide buffer (0.05 M, pH 9.0). The absorbance was measured at a 232 nm wavelength.

For calculating relative enzyme activity (U) in each of the assays above, the absorbance changes of the extracted solution were recorded over 1 min.

## 2.6. Determination of ROS and Active Oxygen Scavenging Enzymes Activity in Grafted Plants

The level of ROS and the activity of active oxygen-scavenging enzymes were measured following protocols already described in several reports [36–39]. For those analyses, enzyme extraction buffer (2.5 g polyvinyl pyrrolidone (PVP) in a 100 mL 0.05 M, pH 7.8 phosphate buffer) was prepared prior to plant material processing. Afterwards, samples taken from stems (0.2 g) and roots were individually thoroughly ground with a mortar and pestle and diluted with 2.0 mL extraction buffer in an ice bath. Afterwards, solutions were centrifuged at 12,000× *g* and 4 °C for 30 min and supernatants (extracted solution) were collected for the next experiment.

To determine the content of the superoxide anion (O$_2^-$), the protocol described by Dhindsa et al. [37] was followed. Briefly, 1.0 mL of extracted solution was mixed well with 1.0 mL phosphate buffer (50 mM, pH 7.8) and 1.0 mL hydroxylamine hydrochloride (1 mM) solution. The mixture was incubated at 25 °C for 1 h. Further, 1.0 mL P-aminophenylsulfonic acid (1.0 mmol·L$^{-1}$) and 1.0 mL α-naphthylamine (7 mmol·L$^{-1}$) were added. The absorbance was measured at a 530 nm wavelength.

To determine the content of hydrogen peroxide (H$_2$O$_2$), the method described by Dhindsa et al. [37] was followed. Briefly, 1.0 mL of extracted solution was mixed well with 0.4 mL color-developing solution (containing 100 mL of 0.1 M phosphate buffer, 25 μL of N, N-dimethylaniline, 100 mg of 4-aminoantipyryline), and 0.1 mL horseradish peroxidase (250 μg·mL$^{-1}$). The absorbance was measured at a 550 nm wavelength.

For superoxide dismutase (SOD) activity determination, the method previously reported by Kochba et al. [38] was followed. Briefly, 0.1 mL of previously extracted solution was mixed with 2.6 mL reaction mixture solution (102 mL of 0.05 M, pH 7.0 phosphate buffer; 18 mL of 130 mM DL-methionine (MET); 18 mL of 750 μM nitro blue tetrazolium (NBT); 18 mL of 100 μM EDTA-Na$_2$), and 0.3 mL

riboflavin (20 μM). After a 15 min reaction under lights (4000 lx), the reaction was blocked in the dark for 5 min. The absorbance was measured at a 560 nm wavelength.

To determine catalase (CAT) activity, the method described by Kochba et al. [38] was followed, which consisted of mixing 0.1 mL of extracted solution within a 2.9 mL phosphate buffer (0.05 M, pH 7.8) and 0.1 mL $H_2O_2$ (2% *v/v* concentration). The absorbance was measured at a 240 nm wavelength.

For peroxidase (POD) activity determination, the protocol described by Dalton et al. [39] was followed. Briefly, 0.1 mL of extracted solution was mixed with a 3 mL reaction mixture solution (100 mL of 0.02 M phosphate buffer at pH 6.0, 38 μL of 30% *v/v* $H_2O_2$, and 0.112 mL of guaiacol). The absorbance was measured at a 470 nm wavelength.

To determine ascorbate peroxidase (APX) activity, the method described by Dalton et al. [39] was followed. Briefly, 0.1 mL of extracted solution was mixed well with 2.6 mL phosphate buffer (0.05 M, pH 7.0, containing 0.1 mM of EDTA-Na$_2$), 150 μL $H_2O_2$ (20 mM), and 150 μL ascorbic acid (5 mM). The absorbance was measured at a 190 nm wavelength.

For calculating relative enzyme activity (U), the absorbance change value of the extracted solution was recorded over 1 min.

## 3. Results

### 3.1. Susceptible Scion Grafts Showed a Weak Tolerance to R. solanacearum

Disease index (DI) and incidence rate (IR) parameters were calculated according to the disease classification standard (Figure 1). The IR of graft S21/S21 was 8.33%, and the DI was 7.23, showing a high tolerance level. The IR of graft Rf/S21 was 26.67%, and the DI was 22.64, showing normal tolerance. The IR of graft Rf/Rf was 100%, and the DI was 97.08, showing high susceptibility (Table 1). The non-grafted plant Rf exhibited a similar level of susceptibility as the Rf/Rf graft, suggesting that grafting did not enhance susceptibility of the susceptible scion (Rf). The disease tolerance level of the different graft combinations was S21/S21 > Rf/S21 > Rf/Rf, which demonstrated that the tolerance level of grafted rootstock decreased by grafting with the susceptible scion (Rf). Thus, the tolerance of a tolerant rootstock was attenuated with the susceptible scion graft (Rf/S21).

**Table 1.** Tolerance of different grafting combinations to *R. solanacearum*.

| Grafted Combination (Scion/Rootstock) | No. Inoculated Plants | No. Infected Plants | Incidence Rate (IR) (%) | Disease Index (DI) | Tolerance Level |
|---|---|---|---|---|---|
| S21/S21 | 60 | 5 | 8.33a [z] | 7.23a | HT [y] |
| Rf/S21 | 60 | 16 | 26.67b | 22.64b | T |
| Rf/Rf | 60 | 60 | 100c | 97.08c | HS |
| Rf (non-grafted) | 60 | 60 | 100c | 96.25c | HS |

[z] Different lowercase letters indicate a significant difference (*P* < 0.05) according to Tukey's test. [y] HT indicates high tolerance, T indicates tolerance, and HS indicates high susceptibility; the Rf non-grafted plant was the experimental control.

### 3.2. Susceptible Scion Grafts Have a Worse Growth Situation Post Inoculation with R. solanacearum

Plant growth parameters, measured 28 days post-inoculation (DPI) with *R. solanacearum*, were used to investigate the effects of the susceptible scion on rootstock tolerance upon infection with the pathogen. There was no significant difference in root length and root surface area between grafts Rf/S21 and S21/S21 in the non-inoculated *R. solanacearum* (Data S1). However, pathogen-inoculated graft Rf/S21 exhibited a shorter root length and less surface area compared to S21/S21 (Figure 2). This negative effect on plant growth was also observed for plant height, stem diameter, and dry and fresh weight (Figures 3 and 4). Without inoculation, root length, surface area, stem diameter, and plant weights were similar for Rf/S21 compared to S21/S21. Because inoculation with *R. solanacearum* inhibited growth of Rf/S21 plants, growth of the tolerant rootstock was abated by the susceptible scion.

These findings suggest that the susceptible scion (Rf) had a negative effect on the eggplant rootstock post-inoculation with *R. solanacearum*.

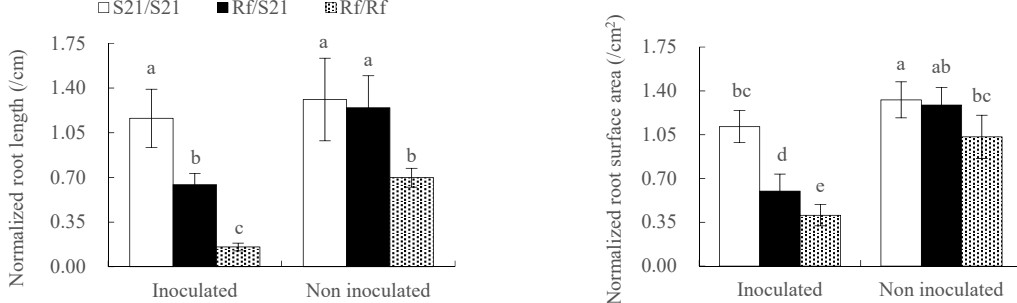

**Figure 2.** Normalized root length and root surface area in different graft combinations. The error bar indicates the standard error, *n* = 30; the different letters indicate a significant difference (*P* < 0.05) across all combinations according to Tukey's test.

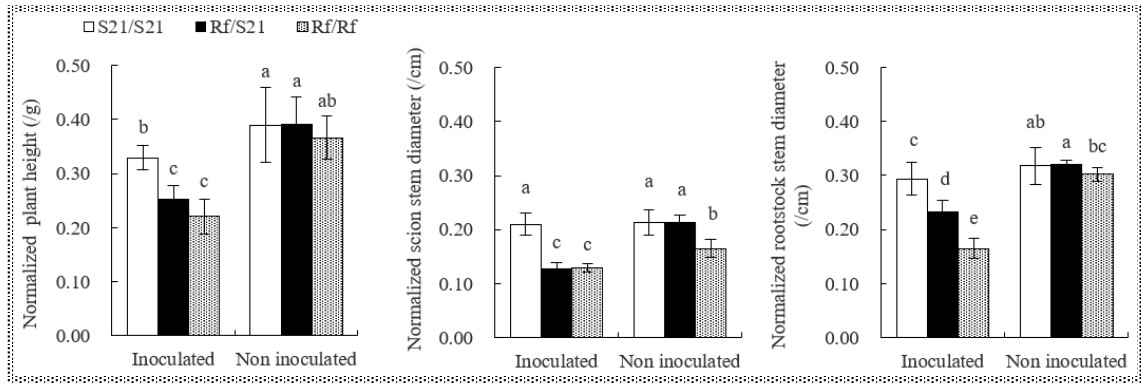

**Figure 3.** Normalized plant weight, scion stem diameter, and rootstock stem diameter in different graft combinations. Note: The error bar indicates the standard error, *n* = 30; the different letters indicate a significant difference (*P* < 0.05) across all combinations according to Tukey's test.

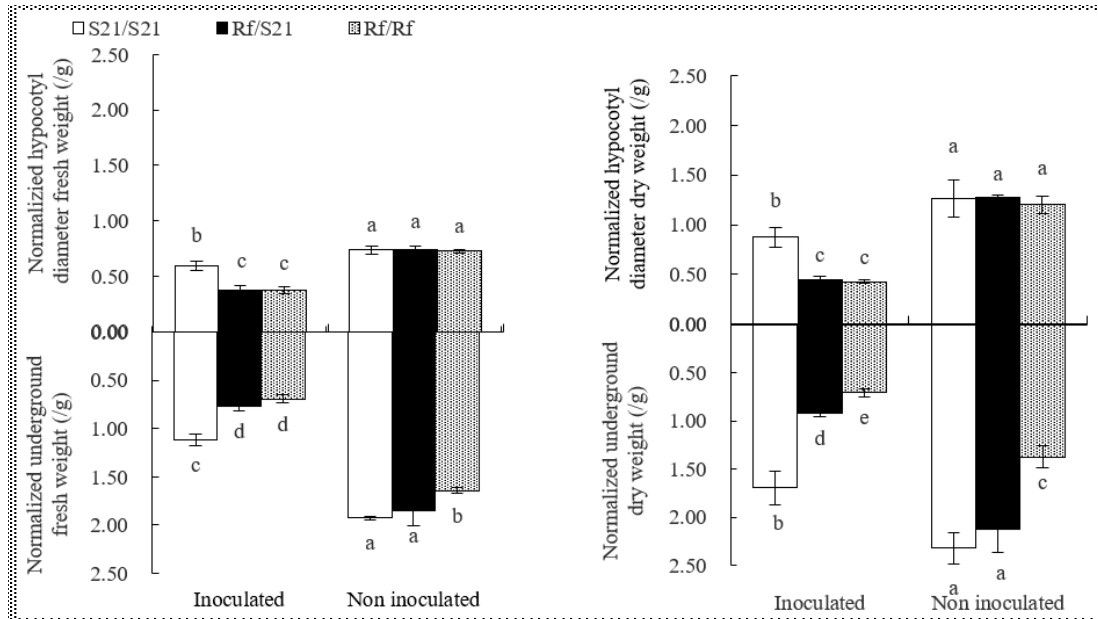

**Figure 4.** Normalized dry weight and fresh weight of plant stems and roots in different graft combinations. Note: The error bar indicates the standard error, *n* = 30; the different letters indicate a significant difference (*P* < 0.05) across all combinations according to Tukey's test.

### 3.3. High Levels of EPS and CWDEs Accumulated in Susceptible Scion Grafts

To further investigate the inhibitory effect of the susceptible scion on the tolerant rootstock after infection with bacterial wilt, the exopolysaccharide (EPS) content and cell wall degrading enzyme (CWDE) activities were measured in the rootstock (Data S2). Rf/S21 produced more EPSs than S21/S21 in both stem and roots. The EPS continuously accumulated in Rf/S21 rootstock over time, reaching a peak at 28 DPI with *R. solanacearum* (Figure 5). The EPS content of the S21/S21 rootstock reached a peak value at 14 DPI with *R. solanacearum* and subsequently declined. In Rf/Rf (high disease index, Table 1), the EPS accumulation of the rootstock was much higher than that in Rf/S21 and S21/S21. Similarly, the CWDE (Cx, PMG, and PMTE) activities of Rf/S21 rootstocks were higher than those in the S21/S21 rootstock, both in stem and roots. These enzyme activities also reached their peak value in the S21/S21 rootstock at 14 DPI with *R. solanacearum* and subsequently declined (Figures 6–8). Overall, the EPSs and CWDEs accumulated in the tolerant rootstock (S21) grafts less than in susceptible scion (Rf) grafts. These results demonstrate that high levels of EPS and CWDE accumulation can be the cause of graft susceptibility upon bacterial wilt infection.

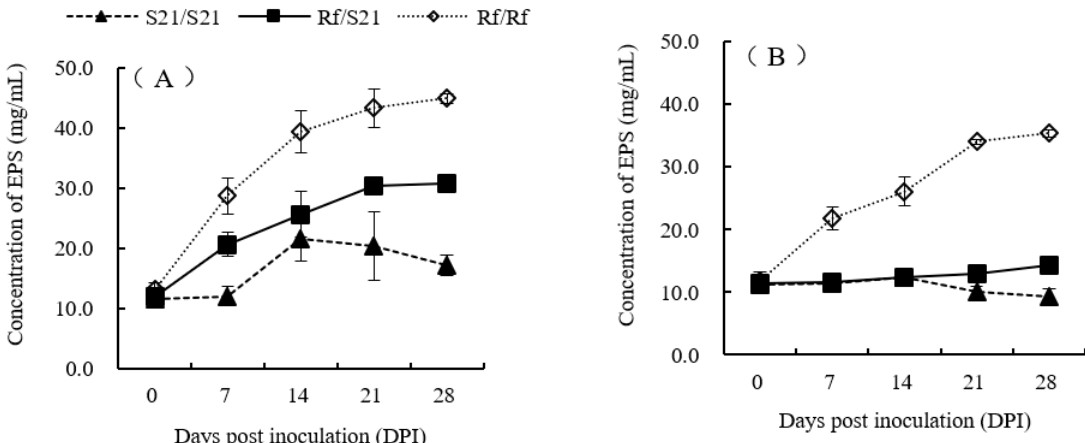

**Figure 5.** Concentration of the exopolysaccharides (EPSs) in the different graft combinations post-inoculation with *R. solanacearum*. Note (**A**) rootstock stems; (**B**) rootstock roots; the error bar indicates the standard error, *n* = 30. See Supplement for mean separations.

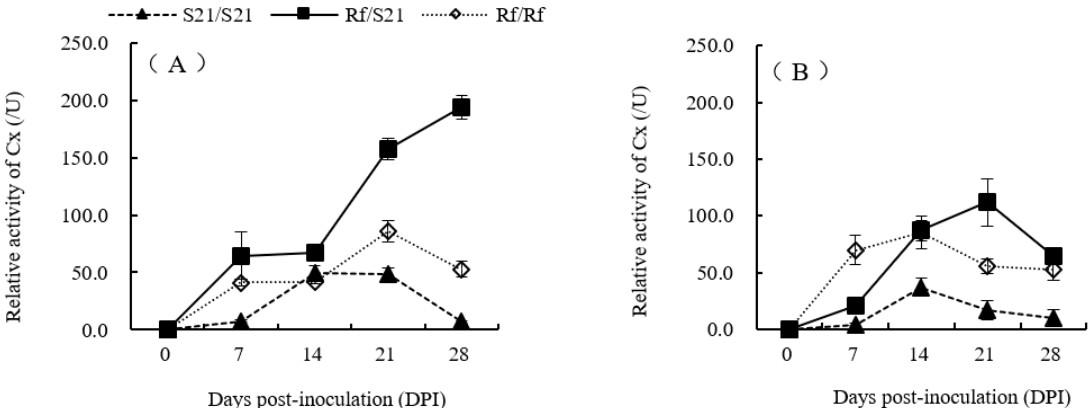

**Figure 6.** Relative activity of cellulase (Cx) in different graft combinations post-inoculation with *R. solanacearum*. Note (**A**) rootstock stems; (**B**) rootstock roots; the error bar indicates the standard error, *n* = 30. See Supplement for mean separations.

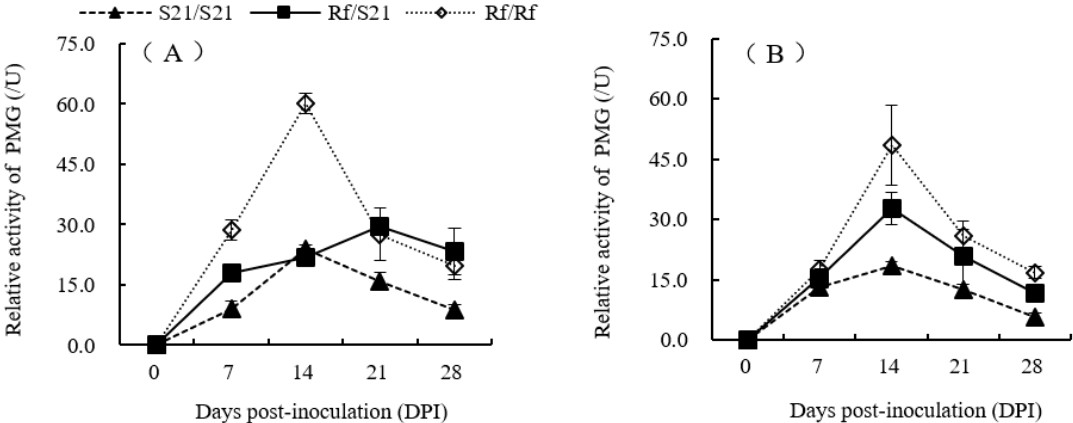

**Figure 7.** Relative activity of the pectin methylgalacturonase (PMG) in different graft combinations post-inoculation with *R. solanacearum.* Note (**A**) rootstock stems; (**B**) rootstock roots; the error bar indicates the standard error, *n* = 30. See Supplement for mean separations.

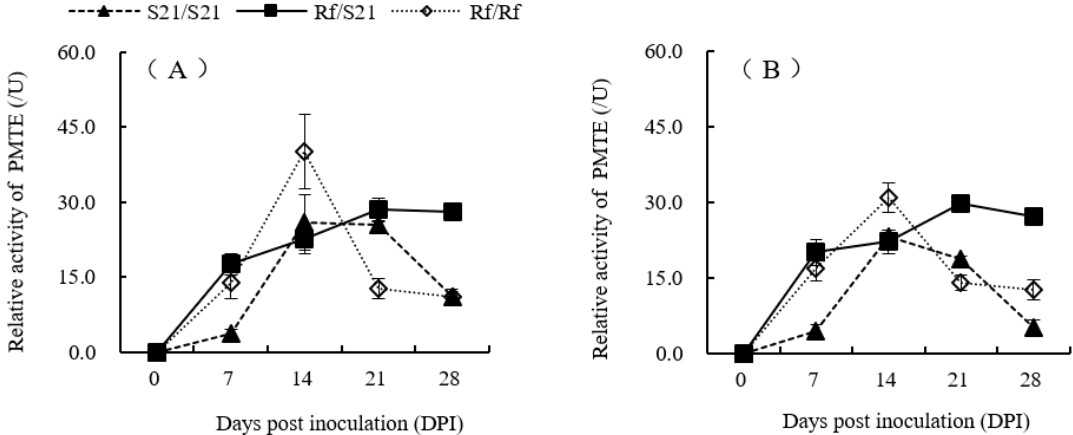

**Figure 8.** Relative activity of the pectin methyl transelimination enzyme (PMTE) in different graft combinations post-inoculation with *R. solanacearum.* Note (**A**) rootstock stems; (**B**) rootstock roots; the error bar indicates the standard error, *n* = 30. See Supplement for mean separations.

*3.4. High Levels of ROS and Active Oxygen Scavenging Enzymes Accumulated in Susceptible Scion Grafts*

In order to further investigate the defense factors of the tolerant rootstock against *R. solanacearum*, the level of ROS (content of $O_2^-$ and $H_2O_2$) and the activities of active oxygen scavenging enzymes (SOD, CAD, POD and APX) were measured in the rootstocks (Data S3). The $O_2^-$ and $H_2O_2$ content in each part of the S21/S21 rootstock were basically maintained at the same low levels, even at different DPIs with *R. solanacearum*, indicating that the ROS maintained their balance in the self-grafted rootstocks (Figures 9 and 10). The Rf/S21 rootstock reached a peak value for $O_2^-$ concentration at 7 DPI with *R. solanacearum*, then this value began to decrease and ultimately became close to the $O_2^-$ concentration of the S21/S21 rootstock at 14 DPI (Figure 9). In addition, the $H_2O_2$ content in each part of the Rf/S21 rootstock was always higher than that in S21/S21 (Figure 10). These results indicate that the susceptible scion (Rf) induced an increased accumulation of ROS in the tolerant rootstock (S21) upon *R. solanacearum* infection. Meanwhile, a high-level accumulation of ROS suggests that the susceptible scion (Rf) reduced tolerance of the rootstock (S21) causing *R. solanacearum* to more easily infect the host plant.

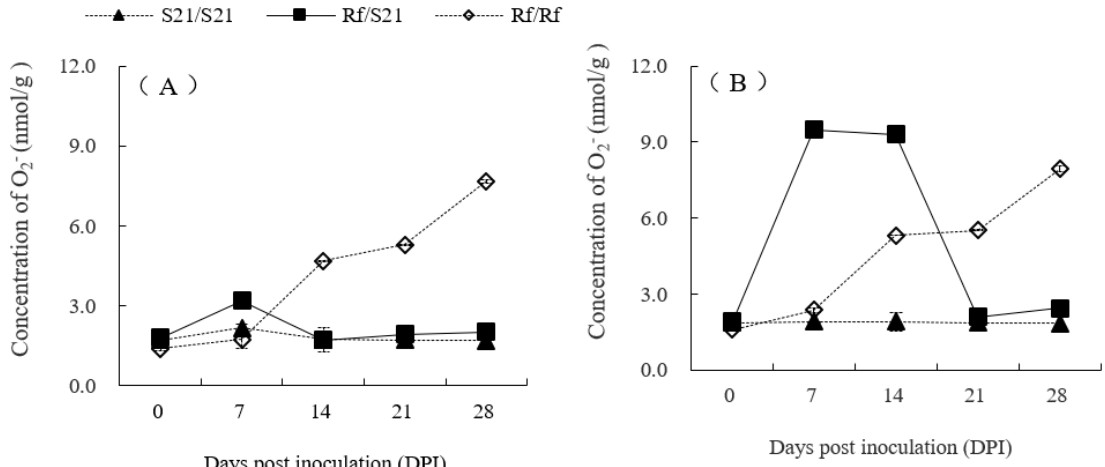

**Figure 9.** Concentration of $O_2^-$ in different graft combinations post-inoculation with *R. solanacearum*. Note: (**A**): rootstock stems; (**B**): rootstock roots; the error bar indicates the standard error, *n* = 30. See Supplement for mean separations.

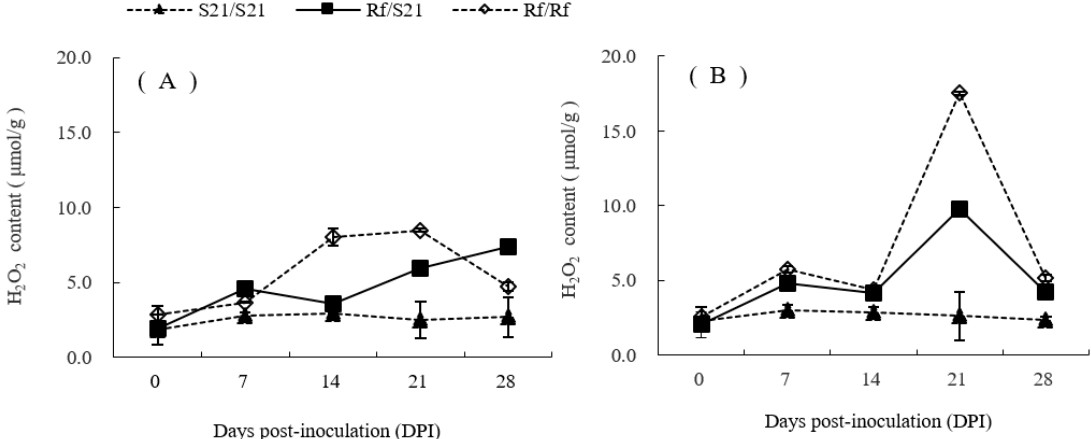

**Figure 10.** The $H_2O_2$ content in different graft combinations post-inoculation with *R. solanacearum*. Note: (**A**): rootstock stems; (**B**): rootstock roots; the error bar indicates the standard error, *n* = 30. See Supplement for mean separations.

In order to alleviate the damage of ROS on plant tissues, plants maintain a relative balance of ROS through active oxygen scavenging enzymes because more ROS is generated in response to pathogen infection. In addition to the SOD activity of the Rf/S21 rootstock (similar to S21/S21), the activity of CAD, POD, and APX were significantly higher in the Rf/S21 rootstock when compared with the S21/S21 rootstock post *R. solanacearum* inoculation (Figure 11). This also suggests that tolerance of the rootstock is affected by the susceptible scion, breaking the balance of the active oxygen scavenging system in the grafted rootstocks by the excess accumulation of ROS and active oxygen scavenging enzymes. The tolerant rootstock damages its own tissues via the imbalance of the active oxygen scavenging system when facing *R. solanacearum* infection in Rf/S21, resulting in reduced growth and further reducing its disease tolerance (Figures 2 and 4). Therefore, continuous infection of *R. solanacearum* in grafted eggplant would induce death in some of the grafted plants, even if they have a high tolerance in their rootstocks (Table 1).

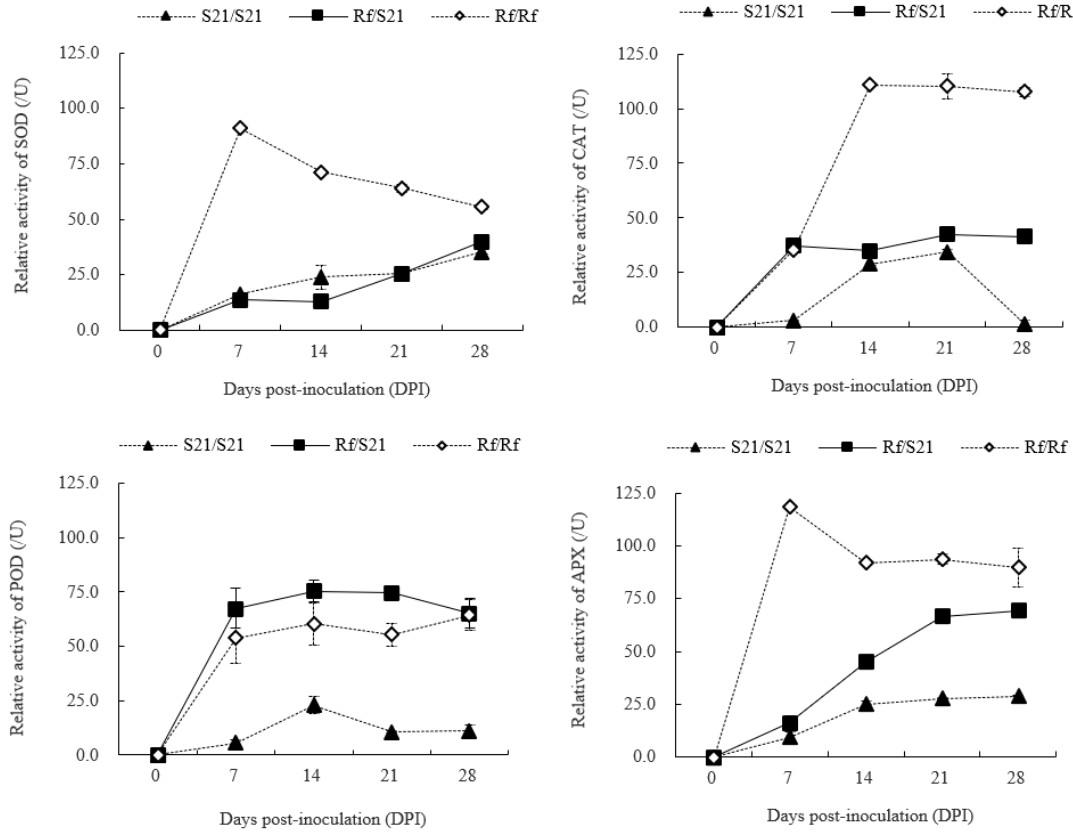

**Figure 11.** Relative activity of superoxide dismutase (SOD), catalase (CAT), peroxidase (POD), and ascorbate peroxidase (APX) of the rootstock roots in different graft combinations post-inoculation with *R. solanacearum*; the error bar indicates the standard error, *n* = 30. See Supplement for mean separations.

## 4. Discussion

A susceptible scion (Rf) on a tolerant rootstock reduced the biological yield of plants to a certain extent when infected with *R. solanacearum*, which corroborates results previously described for other plant species, including pepper (*Capsicum anuum L.*) and tomato (*Solanum lycopersicum*) [40,41]. Now, this phenomenon was documented for eggplant through this study. Currently, more attention has been given to the selection of a tolerant rootstock for grafting, while less attention has been given to the importance of scion tolerance [24]. The effect of the scion upon rootstock tolerance is significant in practical plant production, because susceptible scions cannot achieve their expected economic output due to a reduction in rootstock disease tolerance. Therefore, it will be meaningful to pay attention to the selection of disease tolerant scions in future vegetable grafting processes. The accumulation of EPS and the activity of CWDEs in eggplant rootstocks directly reflect the ability of rootstock plants to successfully react against *R. solanacearum* infection. Although it is not completely understood how these substances affect disease tolerance in grafted plants, the present results showed that *R. solanacearum* secreted massive amounts of EPS and CWDEs in rootstocks at 14 DPI with *R. solanacearum*. Pathogens could be prevented from harming eggplant growth if the inspection and defense work of plants is performed in time (before 14 DPI). Additionally, perhaps more effective EPS and CWDE detection systems could be developed to detect the infection of *R. solanacearum* in time and reduce losses as possible.

In plants, cell-to-cell communication plays a critical role in development, disease tolerance, and responses to diverse environmental stresses. Various types of plant RNA species move from cell to cell (short-range) or systemically (long-range) to potentially regulate whole-plant physiological processes [2,42]. Plant vascular systems are constructed by specific cell wall modifications, through which cells are highly specialized to make conduits for water and nutrients. *R. solanacearum* primarily causes pathogenesis by invading the vascular system, which obviously is involved in systemic substance

circulation and regulation in plants. Plants resist the invasion of *R. solanacearum*, not only due to local tissue reactions, but also due to common systemic regulation. In this study susceptible scions showed a weak disease tolerance. It remains possible that some factors from susceptible scions move into tolerant rootstocks and reduce their tolerance. Recent studies have revealed that CLE and CEP peptides secreted in the roots are transported above ground via the xylem in response to plant–microbe interactions and soil nitrogen starvation, respectively [43]. Thus, plant vascular systems may not only act as conduits for the translocation of essential substances but also as long-distance communication pathways that allow plants to adapt to changes in internal and external environments at the whole plant level [44]. Our research on how susceptible scions influence the tolerance level of rootstocks has led us to propose a process involving the regulation of long-distance signals in plants. There may be a specific interaction between the scion and rootstock, and a grafted plant always maintains the exchange of substances through the xylem and phloem [45], in which disease-related substances (some toxic proteins) in the rootstock are transmitted upwards. If some substances (defense factors) from the scion are conducted downwards over time, the rootstock suffers weakened disease tolerance [46,47].

Although our experiment does not study the exchange or circulation of substances in grafted plants, the direct factors for the weakened disease tolerance in the tolerant rootstock were detected. These factors are directly associated with a decrease in disease tolerance, shown by the accumulation of EPSs and CWDEs in graft tissues (Figures 5–8). Furthermore, through measuring the related balance system of the ROS and active oxygen scavenging enzymes, the reasons for weakened disease tolerance of tolerant rootstocks were explained. The overproduction of $O_2^-$ and $H_2O_2$ would damage plant tissues, while a complex dynamic balance in plants could offset these damages through the actions of SOD, CAT, POD, and APX (Figures 9–11). It is evident that proteins and a range of RNAs can be transported via the phloem, and some of these elements can exert their physiological functions in different plant tissues [44]. Therefore, it could be some unknown substance(s) secreted in the scion, and transport to the rootstock affecting its disease tolerance. In the future, the factors that regulate rootstock tolerance could be studied by big data analysis of RNA-seq to further study the mechanism of the interaction between scion and rootstock.

## 5. Conclusions

A susceptible scion reduced the disease tolerance of a bacterial wilt-tolerant rootstock upon *R. solanacearum* infection. Thus, the susceptible scions showed poorer growth. Moreover, the excess accumulation of EPS and CWDEs in plants with susceptible scions resulted in weakened disease tolerance of the rootstock. Furthermore, the excess accumulation of the ROS and active oxygen scavenging system disrupted the balance of the active oxygen scavenging system in the grafted plants, attenuating the disease tolerance of the rootstock.

**Supplementary Materials:** The following are available online at http://www.mdpi.com/2311-7524/5/4/78/s1, Table S1: Experiment arrangement, Data S1: Plant biomass, Data S2: EPS and CWDE analysis, Data S3: ROS related analysis.

**Author Contributions:** Conceptualization, C.H. and Y.W.; Methodology, Y.Y.; Software, Y.W.; Validation, C.H., Y.W. and Y.Y.; Formal analysis, Y.W.; Investigation, W.Y.; Resources, C.Z.; Data curation, M.N. and C.Z.; Writing—original draft preparation, C.H.; Writing—review and editing, C.H.; Visualization, M.N. and W.Y.; Supervision, W.Y.; Project administration, W.Y.; Funding acquisition, W.Y.

**Funding:** This work was supported by the National Nature Science Foundation of China (No. 31660568) and the Science and Technology Major Project of Guangxi (No. AA17204039-2, No. AA17204026-1).

**Conflicts of Interest:** The authors declare there are no conflict of interest regarding the publication of this manuscript. The funders had no role in the design of the study; in the collection, analyses, or interpretation of data; in the writing of the manuscript, or in the decision to publish the results.

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
