# Peer review of "A Susceptible Scion Reduces Rootstock Tolerance to Ralstonia solanacearum in Grafted Eggplant"

_horticulturae, doi:10.3390/horticulturae5040078_

Round 1

Reviewer 1 Report

horticulturae-571915-peer-review-v1

Title: Susceptible scion reduces rootstock resistance to Ralstonia Solanacearum in eggplant grafts

This manuscript deals with a study performed in eggplant (Solanum melangena L.) that intents to investigate the effect that the scion genotype presents, in terms of resistance to soil-borne Ralstonia solanacearum, on grafted plant performance upon infection with that pathogen. The authors showed that resistance to the soil-borne Ralstonia solanacearum even if used a resistance rootstock, depends on the susceptibility of the scion used for grafting.

The topic is highly interesting and the results described are new and have practical application. The manuscript would merit of publication in horticulturae journal; however, on its present form the manuscript reveals serious flags that turn it not acceptable for publication.

I would ask authors for an additional effort in order to improve it. 

A language revision must be considered.

I suggest Major Revision

Issues to be considered

Introduction

In my point of view the introduction is acceptable. However, depending on the improvement that authors will consider in the discussion section, it could be necessary to complete this section.

Revise the text from line 64 till 67.

Materials and methods

In materials and methods section, the subsection “Plants and Pathogens” must be changed. I can suggest “Plant material and Ralstonia solanacearum inoculum”

Line 93: it is missing the verb “the pathogen was mixed with 30% glycerol…”

Line 94: “used for subculture” change by “used for inoculum subculture”

In line 95: “aqueous solution” means liquid culture medium? If so, please provide composition or at least the reference to that medium.

Lime 98: include composition of “seedling substrate for germination”

Line 99: include reference or the description of the grafting method used (lose joining method).

Line 100: please revise the conditions used. It is not clear if conditions have changed during the experiment. Rephrase like “Grafted seedlings were kept during 12h in a growth chamber under controlled conditions (24-28ºC, 90-95% relative humidity under dark). Further, to promote grafted wound healing, temperature of growth chamber was changed to 20-22ºC for three days.”

Line 102: include the volume of the vessels, the substrate used and the number of plants considered per vessel. The total number of plants per condition is also here missing. Here appears the indication of “after 15 days”, please revise this text, the procedure is not clear.

Line 106: please rephrase this sentence. I would suggest “After inoculation, these seedlings were incubated in a growth chamber for 28 days under 14h photoperiod with 30ºC day/ 25ºC night and 70% of relative humidity. Cultures were watered and fertilized twice a week during the period of the experiment.”

Line 111: It must here be clear the timepoints in which the level of infection was determined. Here the authors point 5 timepoints but in results section this is not clear. On contrary, seems that these parameters were measured only at28 days post inoculation. Please clarify this question and make it clear for readers.

I would suggest to change “days after inoculation” by “days post inoculation (dpi)”

First paragraph must be changed. I would suggest “ Data related with disease symptoms were taken at xxxx days post inoculation (dpi) with R. solanacearum. The level of disease resistance was evaluated 28 dpi following procedure described by Liu et al. (2005). Four different levels were defined: Grade 0 – no obvious symptoms, … The parameters, Incidence rate of infection and disease index (DI), were determined according with the mathematical expressions below.  Five levels of resistance upon bacterial wilt were considered (based on DI parameter, following some reference?): I – immune (DI=0), …”

The statistics made must be better described. I would suggest to make a subsection only for statistical analysis. Please include the software used.

Line 123: Authors give here the indication about the number of plants “a total of 10 plants at each combination”. This means that only 10 plants were analyzed, or have they considered additional repetitions of the experiment? However, in Table 1 authors are giving the indication of the number of inoculated plants as 60 per combination, which is really confusing.

Also in the subsection “Measurements of plant growth characteristics” must be clear the timepoint when data were collected.

Line 137: In subsections related with biochemical parameters, I would request to revise all procedure. Please start by giving the indication about the timepoints considered. Make also clear if samples were collected from stems, leaves or roots in all timepoints. Before starting a new procedure (like in line 148) make a brief introduction to that parameter, otherwise it is very confusing to understand that the procedure described is for determination of a different parameter. Always when the indication of a “buffer” is used authors must describe it composition or at least give a reference to the manuscript where that information is available.

The meaning of reaction solution is not clear understood that “reaction solution” will be used for determination of several parameters but this is not clear. Please rephrase the sentence in line 146.

It must be included the number of replicates considered (biological and technical).

Results

Please include the information regarding the timepoint in line 208.

In Table 1 please consider my previous comment regarding the number of plants.

I would also suggest to change in Table 1 “CK” by “Non-inoculated”. Considering that Non-inoculated were considerd in the same three combinations  as the Inoculated, why are those combinations not considered in the table as they are in the graphs?

The legends of graphs from Fig. 2, 3 and 4 must be changed.In y-axis it must be included the units (cm, mm, …?), and in x-axis it mus be clear that CK were non-inoculated plants. I suggest to change “Inoculation treatment” by “Inoculated” and “CK” by “Non-inoculated”.

Please explain why n=10, it were really used 10 plants per experimental condition? If this is right, in my point of view, the data are not robust enough. In this sense I would request authors to repeat the experiment two more times in order to get the minimum sampling number that can be used for a robust statistical analysis.

Line 258: change “after 0 and 28 days of inoculation” by “previous to inoculation, and 28 dpi”.

Line 262: include “of pathogen inoculation when compared with S21/S21”.

I would suggest to split the graphs of Fig. 4 , making both positive.

Line 316: this last paragraph must be moved to discussion.

Legends from graphs of Fig. 5, 6, 7, 8, 9 and 10 must be changed. Please include in y-axis the units always as possible , and change x-axis to “Days post inoculation (dpi)”. Explain also here why n=3.

Discussion

Discussion is poor, with only four references included. An improvement is mandatory. I would suggest to include studies more recently published; in general authors considered mainly older references, only 3 references are from the last 5 years are included. There are interesting studies regarding, as example circulating DNA between rootstock and scion that could be interesting to include here to support the results achieved.

Author Response

Dear reviewer

Thank you for carefully reviewing this manuscript. According to other reviewers' recommendation, this manuscript has been major revised from abstract to the discussion. Because of plenty of revisions, it not clear to answer your question point by point here. Please see the attachment. Traces of all revisions are also saved in the Word file. Please review it again, thank you very much.

Best regards

Huang Chaokun

Reviewer 2 Report

The present study investigated disease levels in rootstocks with and without grafts with disease-susceptible eggplant scions. Self-grafts of disease-resistant (cv. S21) and disease-susceptible (cv. Rf) cultivars were used as controls, and compared to a graft of Rf scion/S21 rootstock. The phenotypic analysis of the roots of these grafts revealed that the grafting of susceptible scions caused the reduction in the root growth of resistant eggplant species. . These results revealed the significant pathogenic factors of bacterial wilt in the rootstocks of Rf/S21, as evidenced by the higher amounts of exopolysaccharides (EPS) and cell wall-degrading enzymes (CWDEs) excreted from R. solanacearum. Higher accumulation of reactive oxygen species (ROS) and active oxygen-scavenging enzymes were detected in the rootstocks of these Rf/S21 grafts. They concluded that R. solanacearum-susceptible Rf scion grafted onto S21 resistant rootstock resulted in a reduction of tolerance, suggesting that substances from Rf that crossed the graft union into the S21 rootstock were responsible for this phenomenon. Although the overall interest and visibility of this work, some aspects should still be considered to improve the quality and objectiveness of this work.

Abstract is not clear. Abstract needs clear background, objectives, methods, results and conclusion. But the present form of abstract is not clear. Background of the study should be made to very clear. First of all present the background studies about the topic in a manner that set a foundation to understand the research problem with proper reference citations. In “Materials and Methods” majority portion is given without any reference citation. Is it author’s own methodology or extracted from somewhere else? If it is taken from already published literature then provide reference source of that adopted methodology. Rewrite the “Results” with proper subheadings and avoid the extra general and unnecessary information in it. Conclusion” of the study is lengthy. Summarize it in 5-6 lines and provide only the core outcomes of study in it.

Overall,  Figure quality should be improved. Statistical part should be improved. English of the MS needs to be critically improved.

Author Response

(The authors gave the same response as above.)

Reviewer 3 Report

The manuscript ID: horticulturae-571915 entitled "Susceptible scion reduces rootstock resistance to Ralstonia solanacearum in eggplant grafts" by Huang et al. is well structured, although I recommend revision by a native English speaker to correct minor grammar errors. Bacterial wilt is a major disease of vegetables and of broad interest to the readership of Horticulturae. I'm willing to review again a revised version of the manuscript, which should be suitable for publication with minor improvements.

I listed below some suggestions:

Lines 68-73: Please provide some references.

Lines 73-75: Any other reference to cite additionally to Aloni 2010?

Lines 76-79: This is an important background to sustain the major claim of this study. It is very important the authors cite references here.

Line 103: "...physiological indices were determined.". Which ones?

Line 105: Even though inoculation method has been published, it would be nice to have more info, such as for how long the roots where exposed to the bacterial solution or any other parameter important for reproducibility.

Line 121: Provide more details of the type of t-test used. Consider using ANOVA instead of t-test.

Table 1 and Fig. 2: What exactly is contrast check (CK)? Are these grafts that were not inoculated with the pathogen but just buffer? Please clarify.
Review panels alignment in Fig. 3.

Figure 5: Did the authors measure EPS content in non-inoculated plants? The measurement would be informative of the amount of background noise in this experiment.

Figures 6-8: The values shown are "relative". Some more information on which comparisons were made would help interpret the results. Also, a notion of the background noise of such measures in non-inoculated plants would be nice.

Lines 653-654: This sentence might contain an overstatement as a direct link was not tested. I recommend authors tone down this claim or rephrase.

Line 657: Instead of a comma, I suggest a period, dividing the sentence in two to avoid being too long.

Author Response

(The authors gave the same response as above.)

Round 2

Reviewer 1 Report

The changes made by authors in order to improve the manuscript are visible, however, seems that the language revision, which was mandatory, was not considered by authors. Additionally, there are some more points that were not enough improved, which I am listing below.

The present abstract is not clearly written; authors must be aware that they must here include the main objective of the research, the methods used to reach the goals and highlight the most important results. Materials and methods section is still incomplete and very confusing, and statistical methods are not enough explained. The quality of the figures is not good, and the legend of graphs were not all completed. Discussion still requires an improvement, and references must be formatted.

I reject.

Author Response

Dear reviewer

Thank you for reviewing again. Although you have rejected my manuscript, I still appreciate your kindness and patience review.  According to the reviewers' comments, this paper has been polished by MDPI English editor, it seems no problem in English writing now.

Best regards

Huang Chaokun

Reviewer 2 Report

Requested corrections has been completed. No statistical analysis for some figures. Statistical analysis needed all graphs.

Author Response

Dear reviewer

Thank you for reviewing again. According to the reviewers' comments, this paper has been polished by MDPI English editor, it seems no problem in English writing now. And I have include number of replicas for some biochemical assays performed as much as possible.  Besides, about the 3) The authors are interested in rootstock health, but are measuring symptoms (0 to 5 scale used in Figure 1 and Table 1) in the scion. That is a really methodological flaw, because we cant find any reference paper which can directly guild us judge bacterial wilt by rootstock, if we cut the rootstock of plant it would affect next experiment, so we thought using the symptoms of scion may be a suitable chose.

Best regards

Huang Chaokun

Reviewer 3 Report

The manuscript horticulturae-571915 titled "Susceptible scion reduces rootstock resistance to Ralstonia Solanacearum in eggplant grafts" by Huang et al was improved during revision but it is still below Horticulturae standards of quality. The main points to be improved are:

1) Text must be revised by a native English speaker for grammar and clarity.
2) Include number of replicas for all biochemical assays performed.
3) The authors are interested in rootstock health, but are measuring symptoms (0 to 5 scale used in Figure 1 and Table 1) in the scion. This seems to be a methodological flaw. Please comment.

Author Response

Dear reviewer

Thank you for reviewing again. According to the reviewers' comments, this paper has been polished by MDPI English editor, it seems no problem in English writing now. Besides, the statistical analysis has been added, we are thinking some statistical analysis are easy and usual method that can skip the explanations in details.

Best regards

Huang Chaokun

Round 3

Reviewer 1 Report

horticulturae-571915 Revised Version 9.27

Title: The susceptible scion reduces rootstock resistance to Ralstonia Solanacearum in eggplant grafts

In my point of view there are still several points that must be improved before acceptance.

One important issue that I would like to see revised in the manuscript is related with the use of the terms resistance and tolerance. I recommend to the authors to have a look on the manuscript doi:10.3390/ijms19030810 and other references to better define these two terms, and correct their use within the manuscript.

In attachment I am sending the document with my comments included.

I agree with major revision.

Author Response

Dear reviewer

Thanks for your careful review and valuable comments, my manuscript has been major revised again according to your proposal. In additional,about the terms ‘resistance’ and ‘tolerance’ we finally decide using ‘tolerance’, the reason as follow:

About the rootstock germplasm resources, we usually use ‘resistant rootstock’, but disease resistance and disease tolerance were both usually used at research analysis in the published paper, and ‘tolerance’ was more use in recent years. We very agree with your analysis of the terms of resistance and tolerance.

Best regards

Huang Chaokun

Round 4

Reviewer 1 Report

I have included all my comments in the manuscript.

Author Response

Dear reviewer

Thanks for your careful review and valuable comments, my manuscript has been major revised again according to your proposal. In addition, we answer some of your questions as follow:

1.) The “enzyme solution” indicates extracted supernatants in experiments, because they contained CWDEs and Oxygen Scavenging Enzymes, for understanding clearly, we use “extracted solution” instead of them.

2.) About the “units”, we have added the enzyme activity unit (/U) to all graph; the original measure units (cm, cm2, g) have been supplanted after calculating normalized values by log function normalization, so there is no suitable unit in normalized data graphs.

3.) About adding a reference in the discussion part, there is no other references were found. Our experiment is the first to prove that the disease tolerance in eggplant scion affects the disease tolerance of grafted seedlings. Other studies are that rootstocks affect the disease tolerance of grafted seedlings or no in eggplants.

Best regards

Huang Chaokun

Round 5

Reviewer 1 Report

Title: The Susceptible Scion Reduces Rootstock Tolerance to Ralstonia Solanacearum in Eggplant Grafts

General comments to the manuscript:

The changes made by authors in order to improve the manuscript are visible, however, there are still several points that must be revised and improved before final acceptance. I would like to ask authors to take this seriously because some points were already highlighted in previous revision. A revision by a native English speaker must be taken as an important requirement, there are still grammatical and typographical errors that must be corrected.

In my opinion the experimental design is confusing. In the different sub-topics authors use a different number of plants (In 2.3. they write 20 plants x grafted combination x 3 replicates; in 2.4. they report the use of 10 plants x grafted combination x 3 replicates; in 2.5. they  report “Ten plants were selected from each grafted combination at each experimental timepoint.”. Additionally, in some analysis they consider control plants (non-grafted) and sometimes not. It is also not clear if authors used the same plants to evaluate the different parameters, or if they considered different plants. I would ask authors to make a scheme in order to make easier for readers to understand.

Sometime authors write “per experiment” but probably they want to say “per repetition”. I think that all these doubts will be more clear if authors make the scheme.

Figure 1: I have previously made a suggestion to authors to increase the size of figures. I still suggest authors to increase the size of each figure, and to combine all figures in a single one (I made some arrangement in the text to give an idea). A title to the main figure is required as well the bar in each figure.

In the graphs it is still requited the units. Authors write in legends “Normalized” but I don’t understand the meaning and the need of this data treatment. Authors must explain the statistics made and the reason why they followed that normalization strategy.

Please see the document in attachment where I included the main sugestions/corrections.

Author Response

Dear reviewer

Thanks for your careful review and valuable comments, this manuscript has been revised again according to your and supervisor’s advice. In addition, we answer some of your questions as follow:

1.)About you are confusing on “How to know that pathogen showed a good-growth? If a researcher wants to follow this protocol how this parameter can be evaluated?”, we just want to prove this pathogens can be used in our experiment, firstly, this pathogen (R. solanacearum) can grow in the medium without contamination, and then, they have strongly pathogenic in subsequent experiments (susceptible seedlings have died).

2.) About the number of experimental plants, just in case of an accident, we prepared lots of plants and not all of then used in the experiment, to understand clearly, here, I draw a simple chart about the experiment arrangement. (Appendix at the end of the manuscript in attachment)

3.)In our experiment, the comparison between three grafted combinations (S21/S21, Rf/S21, Rf/Rf), we only want to get the result from Rf scion upon S21 rootstock, Rf/S21 is experimental group, S21/S21 is positive control, Rf/Rf is negative control, the Rf (non-grafted) is an additional control in the result of “Tolerance of different grafting combinations to R. solanacearum, expert that, other experiments not containing Rf (non-grafted).

4.) About “Normalized” values, we didn’t use the original biomass measurement value of plants at 28 DPI, firstly, we measure the biomass values of plants at 0 DPI, that is a fixed value, after that, relative biomass values at 28 DPI were measured, these values minus the fixed value at 0 DPI in three experiment (the fixed value we only measure one time no repetition), the final result is normalized values.

5.) Except the content EPS and ROS (O2- and H2O2), the other data from measurement of enzyme actives (Cx, PMG,PMTE, SOD, CAT, POD, APX), about the units of enzyme actives (U),we through measuring absorbance change within 1 min to calculate them, that is common method in the experimental textbook so we didn’t describe in detail in this manuscript. Here, I will explain how to calculate enzyme activities as follow:

One unit of enzyme activity with A change of 0.01 per minute,

U = [∆A / (0.01 × t)] × D

In the formula: U is the enzyme activity; â–³ A is the change of absorbance during the reaction time; t is the reaction time (min); D is the dilution factor, that is.

6.)About the review “It is not clear why the present research is the first one demonstrating the influence of scions on plant tolerance if two previous reports have already stated that dependence.” Because our experiment on the eggplant, they were work on pepper, its different plant, so the eggplant and the bacterial wilt in our experiment is the first time to report.

7.) Please don’t combine all figures in a single one or change their size, this manuscript must be proofreading and modified to a suitable formation by the journal editors or English editors before publish, they need the original charts and figures. So I change them again.

8.) About the English Proofreading,I have pay MDPI for reviewing English writing if there is no other problem in this manuscript it will be sent to the journal to execute English proofreading again.

Best regards

Huang Chaokun

This manuscript is a resubmission of an earlier submission. The following is a list of the peer review reports and author responses from that submission.

Round 1

Reviewer 1 Report

 The Authors gave a sound presentation of the rather complex effects of the scion on the behavior of resistant rootstocks. Nice achievement !

Author Response

Dear reviewer

Thank you for agreeing with my research, according to the other two reviewers' comment, I have to make a major rewrite of this paper.

I will submit my research paper again after rewriting.

Best regards

Reviewer 2 Report

This work examines factors effecting the disease resistance of grafted eggplants. The choice of root stock and scion are investigated along with several pathogen and host markers following pathogen inoculation. A total of 4 pathogen markers (EPS, and 3 CWDEs) and 6 plant markers (2 active oxygen species and 4 enzymes) are tracked with the goal of elucidating resistance of root stock and scion combinations to bacterial wilt. The work points to attenuated resistance of root stocks with susceptible scions.

Comments on the work:

Table 1 includes resistance data based on wilting characteristics. It would be useful to include the number of plants (out of the 60) falling into each of the Grade 0-4 levels. The designations of significant difference at the 0.05 level are provided, but the tests used to define this are not stated. The statistical protocol should be clearly elucidated in the methods section of the work. It is also not clear how CK differs from Rf/Rf and this should be expanded on - the parenthetical aid under CK, "seedling plant of Rf", is not helpful to this reviewer.

In Table 1 the days after inoculation are not indicated, whereas the methods section points out the numbers were evaluated for days 0, 7, 14, 21, and 28 days (line 112).

Figures 2-4 include axis titles that are probably incorrect - the numbers do not represent % changes, but more likely fractional changes. Better titles might be "Normalized root length" instead of "Growth rates of total root length (%)". The titles in ALL figures should also avoid using "growth rates" since no real rate data is typically presented. Figures 1 and 2 seem to be juxtaposed.

In Figures 2-4, the statistical tests need to be clearly identified - are they ANOVA tests or pair-wise t-tests, are the tests compared to a control, and how were the data actually treated?

Figure 2 indicates n=3, but the methods section notes that n=10 samples (line 123) were taken. Which is correct?

The relative height of the different root/scion combinations (defined in lines 134-135) will be different even in the absence of pathogen inoculation. Some comment on that difference is noted in lines 262-264, but the comment does not extend to the Rf/Rf combination. The control data in the absence of pathogen inoculation is needed to understand the real effect associated with pathogen presence and this should be provided.

Figure 4 is a clever figure showing above ground and below ground data. The statistical tests need to be explained, however.

Figure 5 shows EPS content of root stock stems and roots. The title "Concent ..." should be changed to Concentration or Content; whichever is intended by the authors. The level seems high in that 10% of the weight of the rootstock is EPS even though the rootstock was just inoculatede at time t=0 (see graph 5). Do the plants have a background EPS level or is this expected due to a high inoculation dose, or because of the sampling protocol perhaps? One answer would be to clarify what the % number is relative to.

Figure 5 indicates some statistical differences coded by "different letters" but no such codes are indicated. Same for Figures 6-10.

From lines 562 and forward, the notation O2- should be consistently used. The text contains some superscript and some subscript combinations.

The discussion section (lines 706-733) does not sufficiently elaborate the significance of the changes in ROS, CWDEs, and oxygen-degrading enzymes in better understanding the mechanism of attenuated root-stock resistance with susceptible scions. The statement "... the reasons for the weakening of disease resistance of resistant-rootstock were further analyzed" is not informative and should be expanded or eliminated.

Author Response

Dear reviewer

Thank you for agreeing with my research and for giving me a valuable suggestion for the paper revision, according to the other reviewers' comment, I have to make a major rewrite of this paper.

I will submit my research paper again after rewriting.

Best regards

Reviewer 3 Report

The reviewer started to review the presentation and reached line 152 of the raw manuscript before concluding that a major rewrite was needed.  Examples of suggestions to improve the manuscript and make it more clear are provided below.  The review was then refocused on content.  Before any further evaluations of the work are warranted, the authors must add a better explanation and description of the experimental design used and the statistical methods that were applied.  From the results that are presented, this reviewer believes that the results are compelling and may be worthy of publication.  Without an explanation of how plants were organized, replication in space and time, blocking methods, etc., it is not possible to fully evaluate the veracity of the results.  This reviewer encourages the authors to provide these details in the manuscript and to re-submit for review. 

Specific comments on manuscript content and statements:

L. 10-11)  “…that scion genotype can affect the level of tolerance to soil-borne pathogens in disease resistant eggplant (Solanum melongena L.) rootstocks”. 

L. 11-12) “The present study investigated disease levels in rootstocks with and without grafts with disease-susceptible eggplant scions.”

L. 13-14)  “Self-grafts of disease-resistant (cv. S21) and disease-susceptible (cv. Rf) cultivars were used as controls.”

L. 23-24)  “It was concluded that the R. solanacearum-susceptible Rf scion grafted onto the S21 resistant rootstock resulted in a reduction of tolerance, suggesting that substances from Rf that crossed the graft union into the S21 rootstock were responsible for this phenomenon.”

L. 27-28)  “Bacterial wilt is a soil-borne disease of eggplant (S. melongena) caused by Ralstonia solanacearum E.F. Smith which usually….”

L. 29-30)  “Bacterial wilt is a disease in which the pathogen enters the host through the roots and invades the vascular system (Zhang et al., 2009).”

L. 32)  “The pathogen also rapidly proliferates…”

L. 34-37)  “Regarding the pathogenic mechanism of bacterial wilt, Researchers have revealed that R. solanacearum can produce a large number of exopolysaccharides (EPS) while growing in the xylem that can block the xylem and hinder the transport of water (Wang et al., 36 2008).”

L. 37-39)  “Since EPS block water flow through the perforation plates of vascular lobules, resulting in wilting symptoms, EPS are an important bacterial wilt disease factor  (Yang et al., 1994).”

L. 39)  “Meanwhile, R. solanacearum secretes some cell wall-degrading enzymes (CWDES) during host infection,…”

L. 42)  “The role of Cx is a hydrolase that mainly degrades cellulose…”

L. 43-44) “…the role of PMG is to hydrolyzes the pectin sugar bonds, and the effect 43 of PGTE is to breaks the glycosidic bond through the beta elimination reaction…”

L. 45)  New paragraph.  “R. solanacearum pathogens cannot be directly killed or eliminated from plants once the infection cycle has started.”

L. 45-47)  “Therefore, 45 some defense factors are induced to enhance plant resistance, including reactive oxygen species 46 (ROS) superoxide anion (O2-) and hydrogen peroxide (H2O2).”  Reviewer is confused; are these host resistance responses?  Clarification needed.

L. 50-51)  “..accelerates the senescence and death of plants, and ultimately inhibits plant growth (Bowie et al., 2007).”

L. 64-66) “If these enzymes accumulate to excess, damage to plant cells will result (Schutzendubel, et al., 2002).”

L. 68)  “…approach to control bacterial wilt in eggplant.”

L. 69-70)  “Investigations on the interaction of rootstock and scion…”

L. 72)  “…of rootstock on scions.”

L. 73-74)  “…status, water metabolism, and photosynthesis on the scion, but can also increase the resistance or tolerance of the scion to disease, resulting in higher yield (Aloni et al., 2010).”

L. 74-77)  “Although a disease-resistant rootstock can generally improve the resistance of a susceptible scion, the disease resistance of grafted seedlings is generally lower than that of rootstock self-root plants (unclear to reviewer what this is), which also indicates that a susceptible scion may also cause a resistant rootstock to be less resistant.”

L. 77-79)  “However, no research has been reported on the reciprocal combination of scion and rootstock.”

L. 81)  “…explore the resistance of a resistant eggplant rootstock to bacterial wilt caused by R. solanacearum.”

L. 81-82)  “Furthermore, this study demonstrates that a susceptible scion…”

L. 83-85)  “which is mainly through detecting the EPS content 83 levels and cell wall-degrading enzymes activity, and the balance between ROS accumulation 84 and active oxygen-scavenging enzymes.

L. 88)  “The experimental eggplant seedlings genotypes were…”

L. 92-94)  More details about how the pathogen was maintained and handled are needed here

L. 96)  “Grafting Methods, Cultivation of grafted seedlings, and inoculation with R. solanacearum

L. 99-100)  “These grafted seedlings were 99 kept overshadow (explanation needed; is this under shade conditions?  A better description of illumination is needed) under a daytime temperature of 25-28°C,…”

L. 101-103)  “Grafted seedlings were transplanted into larger culture vessels after 15 days, and physiological indices were determined.”

L. 105)  “…using the root damage-root dipping inoculation method.”  Authors should cite an authority in the literature.

L. 107-108)  “…light (7,000 lux) was for 14 hours per day at 30°C, and the dark period was for 10 hours at 25°C; relative humidity was constant at 70%.”

L. 108-109)  “Unified water and fertilizer management were applied during the cultivation.”  It is unclear what this refers to; more detail is needed on water and nutrient management methods.

L. 113-114)  “…the level of disease resistance was evaluated according to Liu et al. (2005).”

L. 124-125)  “The biomass of the upper and underground part of each plant 124 was determined.” How?  By fresh weight?  By dry weight?  More specificity needed.

L. 128-129)  “The diameter of the rootstock referred to the diameter of the tap root at approximately 0.5 cm below the graft interface,…”  The reviewer believes that “tap root” is the correct botanical description of what is being measured.  

L. 131)  “…analyzed using a WinRHIZO 2009c.” More detail needed; what is this entity?  Software?  Another instrument

L. 132)  “…measured using a thousand-digit electronic balance.”  The reviewer believes the authors are specifying a balance with a resolution of .001 g?

L. 138-139)  “Fresh samples consisted of 0.2 g of stems and roots that were macerated in a pestle…”

L. 141)  “The sample solution was obtained by diluting to the mark (this needs to be explained in more detail) with distilled water.”

L. 142-143)  “Then, 11.78 g of NaCl was dissolved with in 0.05 mol·L -1 pH 5.5 acetic acid-sodium acetate (concentration?) buffer, and diluted to 250 ml with extraction buffer.

L. 143-146)  “Afterwards, 0.2 g of fresh sample was obtained, 1.6 mL of extraction buffer was added to grind into a homogenate in an ice bath, centrifuged at 4°C for 20 minutes at 12,000 × g, and the supernatant was collected as the enzyme solution.” Is this redundant with what was presented in l. 138-143?

L. 148-152)  “According to the phenol-sulfuric acid method, 0.5 mL of the sample solution and 1.5 mL of distilled water were mixed in a test tube, 1.0 mL of 90 g·L-1 phenol solution and 5 mL of concentrated sulfuric acid were added to the test tube, and this was left for 30 minutes at room temperature for reaction. The absorbance of the reaction solution was measured at a wavelength of 485 nm.”  Is this the method of Cao et al. (2007)?  If so, it can be removed. 

Author Response

Dear reviewer

Thank you for giving me a valuable suggestion for the paper revision

I will submit my research paper again after major rewriting.

Best regards